Are giant clams (Tridacna maxima) distractible? A multi-modal study

Doyle Ryan
Kim Jonathan
Pe Angelika
Blumstein Daniel T. marmots@ucla.edu
Department of Ecology and Evolutionary Biology, University of California, Los Angeles , Los Angeles , CA , United States of America
Reimer James
Electronic publication date: 2020 Oct 5
Publication date: 2020
Volume: 8
Electronic Location ID: e10050
Received 2020 May 15; Accepted 2020 Sep 6
Copyright: ©2020 Doyle et al.
Copyright year: 2020
Copyright holder: Doyle et al.
License: This is an open access article distributed under the terms of the Creative Commons Attribution License, which permits unrestricted use, distribution, reproduction and adaptation in any medium and for any purpose provided that it is properly attributed. For attribution, the original author(s), title, publication source (PeerJ) and either DOI or URL of the article must be cited.
License URL: https://creativecommons.org/licenses/by/4.0/

Keywords: Multimodal risk assessment, Giant clam, Multisensory risk assessment, Anthropogenic noise

Funding: UCLA Department of Ecology and Evolutionary Biology This work was supported by the UCLA Department of Ecology and Evolutionary Biology. There was no additional external funding received for this study. The funders had no role in study design, data collection and analysis, decision to publish, or preparation of the manuscript.

==============================
To properly assess risk, an animal must focus its attention on relevant external stimuli; however, attention can be reallocated when distracting stimuli are present. This reallocation of attention may interfere with an individual’s ability to effectively assess risk and may impede its response. Multiple stimuli presented together can have additive effects as distractors, and these include stimuli in different modalities. Although changes in noise and water flow are detectable by some bivalves, this has not been studied in the context of risk assessment or distraction. We experimentally exposed giant clams (Tridacna maxima) to changes in water particle movement through underwater sound (motorboat noise) and increased water flow to determine whether these stimuli, individually or together, modified risk assessment or caused distraction. We found that clams responded to sound, flow, and their combination by increasing frequency of mantle retractions (a potential anti-predator response) when exposed to a stimulus. Sound alone did not change risk assessment in either the latency to close or to reemerge following closure. However, when exposed to both stimuli simultaneously, clams increased their latency to close. We suggest that clams perceive sound and flow in an additive way, and are thus distracted. Interestingly, and uniquely, clams discriminate these multimodal stimuli through a single sensory modality. For sessile clams, anthropogenic noise is detectable, yet unavoidable, suggesting that they be especially vulnerable to marine noise pollution.

Introduction

Most animals experience some form of predation risk in their lifetimes (Lima & Dill, 1990). Individuals must be able to properly assess risk of predation, weighing the costs and benefits of either remaining or fleeing (Caro, 2005). Flight or other forms of protection are often associated with large energetic costs; individuals must expend energy to flee and lose out on potential energy that could have been obtained through feeding (Ydenberg & Dill, 1986; Caro, 2005).

Attention is the process of filtering out irrelevant stimuli from the environment in order for the individual to focus on biologically relevant stimuli (Dukas, 2004). Attention to tasks and external stimuli is limited, divisible and is sensitive to different modalities (Dukas, 2004; Chan et al., 2010; Blumstein, 2013). External stimuli can interfere with risk assessment by both masking biologically important signals (Zhou, Radford & Magrath, 2019) and distracting individuals, thus preventing them from focusing on avoiding predation (Rabin, Coss & Owings, 2006; Karp & Root, 2009; Chan et al., 2010). Distraction can also impair performance on other relevant tasks (Maes & Groot, 2003), potentially reducing feeding and reproductive opportunities. External acoustic and visual stimuli are commonly studied as distractions that impede anti-predatory responses in vertebrates (Grueninger & Pribram, 1969; Dukas, 2004; Wale, Simpson & Radford, 2013). However, there is less literature on distraction in invertebrates. A growing literature on giant clams’ (Tridacna maxima) risk assessment shows that they respond to both visual and tactile threats (Stasek, 1965; Wilkens, 1986; Land, 2002; Soo & Todd, 2014; Johnson et al., 2016; Dehaudt et al., 2019).

However, bivalves assess risk in other modalities as well. They alter their anti-predator behaviors based on different chemical and auditory stimuli (Neo & Todd, 2011; Castorani & Hovel, 2015; Peng et al., 2016). They possess labial palps on their mantle, gills and siphon that can detect and respond to local water particle accelerations (Peng et al., 2016; Nedelec et al., 2016). Their palps also allow them to sense pressure changes from underwater sound waves and bivalves have been documented to significantly react to changes in underwater acoustic stimuli (e.g., Mosher, 1972; Peng et al., 2016; Shi et al., 2019; Charifi et al., 2018; Roberts et al., 2015).

Anthropogenic noise is one such external stimulus that could distract bivalves. Marine anthropogenic noise has been steadily rising over recent years due to shipping, motorboat activity, and pile-driving (Hildebrand, 2009). There remains an overall lack of study regarding the detrimental effects of noise with respect to invertebrates (Morley, Jones & Radford, 2014) and bivalves especially (Peng et al., 2016; Shi et al., 2019; Charifi et al., 2018; Roberts et al., 2015; Solan et al., 2016). Bivalves exposed to anthropogenic noise increase bioaccumulation of toxic metals (Peng et al., 2016), alter their digging, sediment uptake and valve closure patterns (Shi et al., 2019; Solan et al., 2016; Roberts et al., 2015) and experience reduced growth (Charifi et al., 2018). However, these studies did not look specifically at how noise potentially distracts bivalves from performing biologically important tasks. This research is critical because while mobile animals can respond to noise pollution through movement and migration away from areas of high intensity (Hirst & Rodhouse, 2000; Slabbekoorn et al., 2010), sessile marine invertebrates lack escape as an option. Since these species may have important roles in their community (e.g., Jordan & Valiela, 1982; Carballo & Naranjo, 2002; Neo et al., 2015; Solan et al., 2016), we need to know more about how they may be negatively impacted by anthropogenic stimuli. The little research that has been done calls for in situ testing and further understanding of the distracting effects of anthropogenic noise (Aguilar de Soto et al., 2013; Buscaino et al., 2016).

Acoustic stimuli propagate waves that oscillate water particles without creating a clear directional flow (Nedelec et al., 2016); however, these directional changes in underwater flow could also distract bivalves. The effect of changes in water flow has been investigated in bivalves in terms of how it influences their feeding (Levinton, 1991; Pilditch & Grant, 1999). However, we are aware of no studies on changes in flow as a potential bivalve distractor. Changes in water flow could indicate an increased potential for feeding; any increase in underwater velocity could bring more edible plankton in the way of filter-feeding bivalves (Levinton, 1991) thereby altering bivalves’ risk versus reward perception. Flow changes could also indicate changes in threat level because it is one of many factors naturally associated with the presence of a moving predator (McHenry et al., 2009).

Giant clams may commonly experience both stimuli in a multimodal way. Multimodal assessment may provide more information about threats and hence reduce uncertainty about risk leading to an increase or decrease in anti-predator responses (Munoz & Blumstein, 2012). Most multimodal research has explored the additive effect of multiple stimuli as a multisensory experience (Munoz & Blumstein, 2012; Munoz & Blumstein, 2019). Whereas a combination of sound and flow will indeed be considered multimodal, giant clams experience changes in water particle motion through a single sensory channel: their palps. If clams respond differently to multimodal stimulation compared to unimodal stimulation, then this will indicate that giant clams could modify their anti-predator behavior in a multimodal way and that this was achieved by a single sensory pathway binding multimodal stimuli. Specifically, the presence of additional stimuli in the form of sound or flow could potentially distract giant clams from the presence of a predator, causing a delay in how fast they retract their mantle.

We asked whether changes in water particle motion due to unimodal sound, unimodal water flow, or a multimodal combination of the two would distract giant clams or modify their risk assessment. When perturbed, giant clams will partially or fully retract their mantle as a potential anti-predatory response (Johnson et al., 2016; Dehaudt et al., 2019; Land, 2002; Wilkens, 1986). Thus, we used the latency to fully retract their mantle as an indicator of distraction levels and the latency for the mantle to fully re-emerge from its shell as a measure of perceived risk.

Materials & Methods

Study subjects

We marked giant clams (n = 32) for subsequent study within Gump Reef in Cook’s Bay, Mo’orea, French Polynesia (−17.482215, −149.827079), a marine protected area that has been the site of previous studies of clam anti-predator behavior (Johnson et al., 2016; Dehaudt et al., 2019). Each clam was >5m apart from each other. We measured each clam’s depth and the length of the each clam’s shell. Clams received four total treatments over four separate days (with a planned day off between subsequent treatments) in a Latin square design: control, flow only, sound only, and flow and sound together. We initially planned to conduct all experiments over 8 days, but a large storm generated considerable runoff which filled the bay with sediment and created poor visibility making it impossible to study the clams. Thus, we waited until the water cleared and thus conducted two treatments 3 days later than planned. We refer to the first two treatments as pre-plume and the second two treatments as post-plume. Data were collected in the mornings, except for one day in which data had to collected in the afternoon due to water clarity issues from storm water runoff. Clams were studied under permits issued by the Government of French Polynesia (permit approved on 21 November 2019).

Flow calibration

We used an underwater aquarium pump (1/55 HP, ECO-FLO, Ashland OH) to manipulate water velocity over the clam. Before experimentation, we calibrated flow by video recording (Crosstour CT7000 underwater video camera, Shenzhen, China) the time it took red dye to travel 10 cm through a clear, plastic tube. We measured the velocity to be 0.43 (±0.06 SD) m/s (N = 10 measurements) where the jet exiting the nozzle of the pump was 15 cm away from the opening of the tube.

Sound calibration

We used an Oceanears DRS-8 MOD 2 underwater speaker (North Canton, OH) to broadcast recordings of outboard motorboats collected underwater. Five recordings of motorboat sounds of similar amplitude were obtained from a previous study (Simpson et al., 2016). The five motorboat recordings consist of a research vessel (5-m-long aluminum hull boat with 30 hp Suzuki outboard motors) motoring 10–200 m away from the hydrophone at various speeds, mimicking boat operations that are common in coral reef environments (Simpson et al., 2016).

To determine how the broadcast sounds compared to that of the reef environment, we broadcast and re-recorded our exemplars underwater using a Wildlife Acoustics Song Meter SM2+ (Concord, MA) with a Wildlife Acoustics hydrophone. We first calibrated the hydrophone by playing 45 s of white noise of a known amplitude (90 dB re 20 µPa measured at 1 m) through an Oceanears DRS-8 MOD 2 underwater speaker (North Canton, OH) above water. The white noise was generated in Audacity (version 2.3.3, 2019) and was measured with a Radioshack (33–2055) Digital Sound Level Meter set to weighting A (Radioshack, Fort Worth, Texas) placed 1 m away from the speaker. The white noise was then broadcast underwater 1 m from the hydrophone at 0.5 m depth. We also similarly broadcasted the five motorboat exemplars (89–91 dB re 20 µPa measured at 1 m) against the calibrated hydrophone. In Praat (version 6.1.109, 2020), we calculated the average intensity (minimum pitch = 100 Hz) of the white noise and then adjusted the scale to reflect the intensity measured above water to which we added 61.5 dB to roughly estimate intensity re 1 µPa underwater (National Research Council, 1994; Gausland, 2000). This created a scaling factor that we used to adjust the amplitude of each motorboat exemplar. Using Praat, we calculated the average intensity of our stimuli across exemplars, which was 150.6 dB (±1.03 SD) re 1 µPa.

Stimulus presentation

Clams received four different treatments over four days: control (pump and speaker off), flow only (pump on, speaker off), sound only (pump off, speaker on) and flow and sound (pump and speaker on) in a Latin Square design. For all four treatments, the clam was subjected to the presence of both the pump and speaker, whether on or off, for 60 s before it was perturbed. The pump and speaker were attached to a 1.5 m PVC pipe. The hose from the pump was positioned 15 cm away from the clam while the speaker was 1 m away pointed at the center of the clam’s mantle. This ensured that the clam was subjected to 0.4 (±0.07 SD) m/s of flow and 149.13 dB (±1.03 SD) re 1 µPa. Clams receiving sound or flow and sound treatment were randomly exposed to one of five motor boat noises.

During this 60 s period before perturbation, we counted the number of mantle twitches (partial retractions) as an indicator of perceived threat. The entire length of the clam’s mantle was then rubbed back and forth twice using a pencil’s eraser attached to a different 2 m long PVC pipe that permitted the person to be ≥1 m away. The latency to close was recorded using either Crosstour CT7000 or a GoPro Hero 4 camera. The latency to re-open was measured both in the field using a digital stopwatch, and using the camera. We defined latency to close as the first sign of retraction of the mantle to the point where the clam is fully or almost closed. We defined latency to reemerge as the time from the first contact of the pencil to the clam returning to its previous undisturbed state that the observer has determined prior to the clam being rubbed. If the clam did not fully reemerge within 3 min, we determined reemergence as the point where its mantle was most fully extended. The camera recorded the entire process from the 60 s before perturbation until the observer was confident the clam had fully reemerged. In 16 cases we were unable to confidently measure the latency to re-open in the field. Due to the extremely high correlation between field and camera playback latency times, we estimated these recorded times from field observations using a prediction equation based on the correlation between measurements (estimated time recorded from video = 1.0263 * stopwatch time + 1.9489; R2 = 0.98; p < 0.001, N = 103 measurements to estimate latency to re-emerge).

To reduce experimental interference, we did not record the precise velocity of ambient water flow during the experiments. However, we noted flow as either negligible or noticeable by observing pieces of macroalgae moving underwater after conducting each experimental treatment. If pieces of macroalgae in the water column at the same depth of the clam moved in a single direction, flow was marked as noticeable. We estimated an upper limit range of ambient velocity during testing by recording the velocity at which a piece of macroalgae moved 10 cm through a clear, plastic tube with a GoPro nine separate times during a day with particularly high underwater flow. We estimated this velocity as 0.09 (± 0.064 SD) m/s, which was substantially lower than our experimental velocity. Thus, we infer that our experimental velocity was substantially greater than background changes in flow. 

Statistical analysis

Prior to analysis, we plotted histograms of variables to check for normality and log 10transformed retractions and latency to re-emerge.

To determine if stimuli were detected by giant clams, we fitted a linear mixed model with the number of retractions as the dependent variable using the lme4 version 1.1-21, (Bates et al., 2014) and lmerTest version 3.1-1 (Kuznetsova, Brockhoff & Christensen, 2017) packages in R version 3.6.2 (R Core Team, 2019). Treatment, trial number, clam size, and plume presence (defined as before or after the disruptive plume) were added as fixed effects. To account for repeated measures, subject ID was added as a random effect. We tested the assumptions of our mixed model by plotting a histogram of residuals (they were approximately normal), plotting a q-q plot (they were approximately linear), and plotting fitted versus predicted values (there was no obvious pattern). We also compared our model to a linear model without subject ID as a random effect to determine if there was a significant difference in individuals’ number of retractions. We estimated marginal means using the MuMIn version 1.43.15 (Barton, 2019) and emmeans version 1.4.4 (Lenth et al., 2020) packages in R and estimated the d-score as a measure of treatment effect size. We plotted estimated marginal means using plotly version 4.9.2 (Sievert et al., 2020).

To determine if multimodal stimuli distracted giant clams, we fitted a linear mixed model as above with the latency to close, and another model with the latency to re-emerge as dependent variables. Again, we added treatment, trial number, clam size, and plume presence as fixed effects and added subject ID as a random effect. Model fit was determined by examining the normality of residuals in a histogram and quantile plot. Again, both models were compared to a linear model without the random effect of subject ID. Estimated marginal means were plotted with standard errors.

We tested for exemplar effects by creating a dataset containing only those where motorboat noise was used and then fitting each of our models with sound exemplar as a fixed factor to determine if any variation in response was explained by the specific exemplar used.

All data and code are contained in Tables S1, S2. Throughout, we set our alpha to 0.05, and thus interpret p < 0.05 as significant.

Table 1 Results from linear mixed effects model on variation in clam response.

This model explains variation in giant clam responses (a, number of retractions in 60 s, b, latency to close, c, latency to reemerge) following a multimodal stimulus experiment. The reference level was no stimulus.

Fixed effects	Estimate ± SE	P	
(a) Number of retractions in 60 s	
Intercept	0.535 ± 0.144	<0.001	
Trial Number	0.0198 ± 0.347	0.569	
After Plume Presence	−0.111 ± 0.077	0.150	
Size	−0.323 ± 1.18	0.787	
Treatment			
Flow	0.280 ± 0.05	<0.001	
Sound	0.169 ± 0.05	0.001	
Flow + Sound	0.277 ± 0.05	<0.001	
Subject ID	0.010	0.101	
(b) Latency to close			
Intercept	2.490 ± 1.125	0.032	
Trial Number	−0.081 ± 0.248	0.743	
After Plume Presence	0.188 ± 0.550	0.733	
Size	19.685 ± 9.437	0.046	
Treatment			
Flow	0.177 ± 0.348	0.613	
Sound	0.054 ± 0.361	0.882	
Flow + Sound	0.749 ± 0.353	0.037	
Subject ID	0.753	0.867	
(c) Latency to reemerge			
Intercept	1.465 ± 0.271	<0.001	
Trial Number	0.012 ± 0.051	0.816	
After Plume Presence	0.368 ± 0.112	0.002	
Size	0.331 ± 2.346	0.889	
Treatment			
Flow	−0.114 ± 0.074	0.127	
Sound	0.014 ± 0.079	0.859	
Flow + Sound	0.024 ± 0.074	0.752	
Subject ID	0.056	0.236	

Results

After controlling for significant random effects explained by individual (Chi-square = 2.83, 1 df, p = 0.007), and non-significant fixed effects of trial number, plume presence and clam size, there were more twitches recorded during a 60 s period when clams were exposed to sound, flow and the combination of the two stimuli compared to the no stimuli treatment (Table 1A, Linear mixed effects model, p < 0.05 for all). There were large effect sizes between the control and each treatment (Fig. 1A, Table S3; d > 0.75). Additionally, there were moderate effect sizes (0.75 > d > 0.4) between unimodal sound and unimodal flow as well as between unimodal sound and multimodal flow and sound (Fig. 1A, Table S3).

Figure 1 A comparison of estimated marginal means of (A) Log10 (number of partial retractions) (B) latency to close (C) Log10 (Latency to reemerge) between all treatments.

Bars indicate one standard error from the means. Letters above bars indicate treatments that are significantly different from each other.

After controlling for significant random effects explained by individual (Chi-square = 10.848, 1 df, p < 0.001), significant fixed effects of clam size, and non-significant fixed effects of trial number and plume presence, we found that the multimodal treatment (combination of sound and flow) significantly increased clams’ latency to close (Table 1B, Linear mixed effects model, p < 0.05) while the other two unimodal treatments (flow only and sound only) had no effect on latency to close when compared to the control (Table 1B, Linear mixed effects model, p > 0.05). There was a small to moderate effect size between multimodal flow and sound and the control; all other comparisons had small effect sizes (Fig. 1B, Table 1B; d = 0.35 and d < 0.3, respectively).

After controlling for significant random effects explained by individual (Chi-square = 22.442, 1 df, p < 0.001), significant fixed effects of plume presence and non-significant fixed effects of trial number and clam size, we found that clams did not significantly change their latency to reemerge after treatments when compared to the control (Table 1C, Linear mixed effects model, p > 0.05 for all). We found moderate effect sizes (0.75 > d > 0.4) between unimodal flow treatment compared to all other treatments as well as the control.

When we tested for exemplar effects, variation in sound exemplars did not explain any variation in clam response.

Discussion

Animals often use cues from a variety of modalities to reduce uncertainty about their environment (Dall & Johnstone, 2002), which potentially permits more precise assessments of risk. Here we used two stimuli, from different modalities, either alone or together, to study anti-predator responses in clams. We found that clams increased their frequency of partial retractions during the 60 s acclimation period after exposure to any treatment. Clam mantle retraction is presumably a discrete anti-predator response used for protection and to scare potential predators with the water expelled from its siphon during retraction (Wilkens, 1986; Land, 2002).

While twitches were not complete valve closures, the twitches were still a physical response that is closely linked to the species’ highest level antipredator response—full mantle closure. Increased anti-predator behavior indicates that motorboat noise may act as a giant clam stressor (Wright et al., 2007). Previous studies have shown the adverse effects of anthropogenic noise on bivalves’ physiological processes such as metabolism and growth rate (Roberts et al., 2015; Peng et al., 2016; Charifi et al., 2018; Shi et al., 2019). Thus, our results add to previous studies in assessing whether the affected physiological processes in bivalves are the result of anthropogenic noise-based distraction.

Sound and flow did not significantly alter clams’ latency to close when tested as separate unimodal treatments. However, when sound and flow were presented together as a multimodal stimulus, clams closed more slowly, indicating that the presentation of multiple stimuli can be distracting. Given that there was no significant effect of each stimulus when presented on its own and that giant clams perceive both stimuli through their palps, a unisensory receptor, we conclude that these stimuli have an additive effect on compromising clams’ attention. These results are consistent with the distracted prey hypothesis in which peripheral stimuli may compromise an individual’s attention from biologically necessary tasks (Chan et al., 2010). Less vigilant prey become more likely targets for predators (Krause & Godin, 1996), and a slower closing time in clams encountering a multimodal stimulus indicates that distracted clams may be more vulnerable to predatory damage.

Clams did not differ in their reemergence time between exposure to treatment and control. However, the flow as a unimodal treatment had a modest, but not significant, decrease in reemergence time from the control (Table 1) and had a moderate to large effect size when compared to unimodal sound and multimodal sound and flow. This modest decrease in reemergence time in response to increased flow around the clam may indicate an increased opportunity to filter feed because there are more sediments and nutrients moving through its environment (Taghon, Nowell & Jumars, 1980; Trager, Hwang & Strickler, 1990; Skilleter & Peterson, 1994; Hardy & Hardy, 1969; Hawkins & Klumpp, 1995). Thus, clams may reemerge faster to take advantage of increased nutrient availability. As a result, increased water flow cannot be considered a neutral distractor. It is rather unique in that it is a biologically significant stimulus, forcing clams to factor foraging opportunity into their decision-making process. Clams must simultaneously manage predation risk and obtain energy through feeding behavior: two potentially exclusive behaviors with their own set of benefits and consequences. If clams reemerge sooner, they could potentially make the mistake of exposing themselves to predators; however, if they stay closed for too long, they are missing out on feeding opportunities. Further studies are needed that explore reemergence time of clams in areas with naturally high levels of flow to determine if they behave differently from clams in areas, like our study site on Gump Reef, with relatively low flow.

These findings further illustrate the negative effects that anthropogenic noise can have on bivalves (Peng et al., 2016; Shi et al., 2019; Charifi et al., 2018; Roberts et al., 2015). These prior studies identified alterations in essential processes such as digging behavior, metabolism, and growth in bivalves (Peng et al., 2016; Shi et al., 2019; Charifi et al., 2018). Of the prior studies that identified negative effects, none, until this one, specifically focused on anthropogenic noise as a distraction. A recent meta-analysis reported anthropogenic noise negatively affects families across the animal kingdom, potentially altering distribution, communication, foraging, and homeostasis in these families (Kunc & Schmidt, 2019). However, species’ responses differ widely based on ecological context and physical constraints (Wright et al., 2007). While mobile species can avoid anthropogenic noise stressors (e.g., Francis, Ortega & Cruz, 2009; Francis et al., 2011; Goodwin & Shriver, 2011; (Nowacek et al., 2007; Weilgart, 2007; Weilgart, 2018), giant clams are, for the most part, sessile and must endure any anthropogenic disturbance. Giant clams play important ecological roles on coral reefs, including water filtration and nutrient sequestration (Neo et al., 2015). Thus, anything that makes more clams more vulnerable to predators may contribute to the loss of these important services that giant clams provide.

Multimodal stimuli are traditionally defined as separate stimuli that are perceived through different sensory channels (Partan & Marler, 2005). Many animals are able to detect multiple stimuli across different modalities and create a unified percept by integrating them (Brown & Magnavacca, 2003; Partan et al., 2010). In contrast to all other studies we are aware of, giant clams perceive multimodal stimuli through a single sensory modality, their palps, but nevertheless integrated these stimuli. It is important to note that giant clams depend heavily on visual cues for risk assessment (Wilkens, 1986; Stasek, 1965). Our manipulations did not create any shading events or cause any visible turbulence, thus we infer that clams were not relying on visual cues to sense the particle motion caused by noise or flow. We therefore suggest that local changes in water flow and changes in underwater noise are indeed multimodal although perceived through a single sensory channel by clams. The clams’ ability to react differently to either unimodal or multimodal treatments of sound and flow is a novel example of how animals can bind multimodal stimulus together through a single sensory pathway to modify their risk assessment.

Sound and flow changes sensed by clams could also be viewed as multi-attribute stimuli. Unlike multimodal stimuli, which are typically perceived through different sensory channels, multi-attribute stimuli are perceived through a single sensory channel. Multi-attribute stimuli are often studied in bioacoustics (e.g., Gerhardt, 1992; Castellano & Rosso, 2007; Naguib & Wiley, 2001; Gerhardt & Schul, 1999; Kershenbaum et al., 2016). In these cases, different components of an acoustic signal, including rate and frequency rises, can send multiple messages or further inform an individual. Previous studies on crayfish (Orconectes virilis) demonstrated that animals can use multiple separate chemical cues perceived through a single olfactory channel (Hazlett, 1999; Bouwma & Hazlett, 2001). Female Italian treefrogs (Hyla intermedia) weight separate components of male mating calls differently when selecting a mate (Castellano & Rosso, 2007). Similarly, great tits (Parus major) assess multiple acoustic attributes of song (Weary, 1990). Female bluehead wrasse (Thalassoma bifasciatum) use multiple visual cues, such as coloration and courtship display, to assess risk (Warner & Dill, 2000). In honeybees, different olfactory stimuli are processed through different clusters of glomeruli, enabling them to process different components of odor through different pathways (Abel, Rybak & Menzel, 2001; Kirschner et al., 2006). The majority of these previous studies have focused on multi-attribute signaling and not strictly multi-attribute risk assessment (Munoz & Blumstein, 2012; Munoz & Blumstein, 2019). In addition, these previous studies focused mainly on terrestrial organisms; more studies are needed on multi-attribute stimulus assessment in marine organisms. We have shown that giant clams can bind together multiple stimuli into one sensory channel, an ability that we suspect is shared with other species.

Conclusions

Giant clams increased their frequency of mantle retractions, a potential anti-predator behavior, in response to motorboat playback, increased flow and the combination of the two. Clams also slowed their closing times when exposed to a combination of sound and flow, indicating distraction through multimodal means. These findings add evidence to the deleterious nature of anthropogenic noise on sessile marine invertebrates and present a novel case of multimodal stimuli processed together through a single sensory channel. We recommend further study of unisensory systems and their relationships to traditionally multimodal stimuli.

Supplemental Information

Table S1 Cohen’s d values calculated from the estimated marginal means

Cohen’s d values were calculated for (A) Number of partial retractions, (B) Latency to close, and (C) Latency to reemerge.

Click here for additional data file.

We thank the Richard Gump South Pacific Research Station for logistical support, Zachary Schakner for the playback equipment, Maud Ferrari for motorboat recordings, John Milligan and Vivian Kim for help with our equipment, and Dana Williams for logistical and statistical help as well as comments on a previous version of this article. The article was improved by addressing the comments of three astute reviewers.

Additional Information and Declarations

Competing Interests

Author Contributions

Field Study Permissions

Data Availability

The authors declare there are no competing interests.

Ryan Doyle, Jonathan Kim and Angelika Pe conceived and designed the experiments, performed the experiments, analyzed the data, prepared figures and/or tables, authored or reviewed drafts of the paper, and approved the final draft.

Daniel T. Blumstein conceived and designed the experiments, analyzed the data, authored or reviewed drafts of the paper, and approved the final draft.

The following information was supplied relating to field study approvals (i.e., approving body and any reference numbers):

This research was conducted under a permit issued by the Government of French Polynesia (permit approved on 21 November 2019).

The following information was supplied regarding data availability:

Data and code are available at Github.

The data sheets are available at:

https://github.com/distractedclams/Clam-Distraction/blob/master/Clam_Multimodal_Datasheet_FINAL.csv.

The R Code is available at:

https://github.com/distractedclams/Clam-Distraction/blob/master/Distracted_Clams_FINAL.R.

The Experimental Trial Video is available at: https://www.youtube.com/watch?v=O8GkVb8yeMo

The Experimental Design is available at: https://github.com/distractedclams/Clam-Distraction/blob/master/Presentation.pdf.

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
