# Peer review of "Are giant clams (Tridacna maxima) distractible? A multi-modal study"

_PeerJ, doi:10.7717/peerj.10050_

## Round 0.1 · original submission · Major Revisions

I have heard back from three reviewers, all of whom have added constructive comments. In particular, reviewer 1 is concerned about aspects of your experimental design, and I can agree with their assessment, so please consider your responses carefully. I look forward to seeing a revised version of your work.

Reviewer 1 ·

Basic reporting

In general, there appears to be a strong emphasis of distraction (caused by stimulus) linked to predation throughout the paper. Though I would differ that distraction cues could affect organisms in other ways such as altered behaviour and stress that leads to reduced physiological functions (i.e. respiration, productivity, etc…). In my opinion, these are potential risks faced by organisms – whether in the form of being eaten up, or reduced growth. Since the authors did not specifically test the effects of predation based on the stimuli offered, I feel that they should include the studies examining other impacts, on top of predation in the Introduction.

Several statements made are not so clear and needs to be addressed. For example, Lines 49-54: “Attention is limited and divisible and because it is sensitive to stimuli in different modalities (Dukas 2004; Chan et al. 2010; Blumstein 2013), external stimuli can interfere with risk assessment by both masking biologically important signals (Zhou, Radford, & Magrath 2019) and distracting individuals, thus preventing them from focusing on avoiding predation (Rabin, Coss & Owings 2006; Karp & Root 2009; Chan et al. 2010).” – the context for attention was not explained earlier, so the link to stimulus is unclear.

Lines 62, 63, 96: The term ‘palps’ alone is a terminology describing appendages found in arthropods. I think the authors are referring to the ‘labial palps’ in bivalves? Please clarify and ensure the accuracy of information between Lines 62-66.

Line 71: Authors mentioned the lack of understanding on how anthropogenic noise affects bivalves, but 5 recent papers were cited (between Lines 71, 72). Suggest the authors briefly explain the outcomes from these studies, which would add value to this section and briefly answering how marine anthropogenic noise distract bivalves. Similarly, Lines 77-80 on the lack of studies should be nuanced, and instead highlight what exactly is the knowledge gaps after knowing that some studies have already been published.

Lines 81-89: Authors broadly suggest that both sound and water flow can alter water particle motion, but there is not much information to elaborate how do sound and/or water flow change water particles movement. Given that it is a major factor influencing how the stimulus reaches the giant clam, I feel more information should be given, with respect to the two stimuli in this study. Also, I don’t think water flow and water particle motion are the same, but the section focuses only on the former point.

Several statements made need to be properly referenced to support the points made. For example, Lines 88-89: “Flow changes could also indicate changes in threat level because it is one of many factors naturally associated with the presence of a moving predator.” – should be provided with a reference.

Lines 90, 96: Is it ‘Giant clams’ and not ‘Clams’? The latter term refers to a general group.

Line 98: Suggest to add “…could modify…”.

Line 103: Suggest to add “as a potential anti-predatory response” as mantle retraction could also be attributed to clearing of irritants such as sediments, or changing light levels (i.e. shadow responses).

Lines 104-106: It is not very clear to me how latency to fully close their mantles is an indication of distraction levels – needs to be better explained here. Another point is, under most circumstances, giant clams do not necessarily extend their full mantle capacity after each retraction. How do the authors ensure full mantle emergence/re-emergence?

Experimental design

Interestingly, the experiments were carried out with wild giant clams where the species is typically embedded in the shallow waters. While the authors had attempted to address the numerous variables under natural reef conditions (such as ambient velocity flow), I think there needs more clarifications from the authors. An illustration of the experiment will be useful.

1) How did the researchers ensure that their presence did not affect the outcomes of their study? For instance, I assume that the researchers had to be in-water to introduce the extended poles and cameras, so there could be some disturbance from researchers.

2) What was the general water depths at which the giant clams were found and/or experiments were conducted? How far apart were the giant clams between each other?

3) What was the general conditions of the day (e.g. sunny, cloud cover, waves, etc…)? Related to 2), both factors can result in shadows and refraction of water ripples that causes the giant clams to become disturbed by these natural factors. My concern is that light changes under natural settings can mislead the outcomes of this study. Even if objects are placed at a distance, the giant clams can perceive light at a good distance based on previous studies.

4) Authors to clarify in text whether all 5 boat sounds were played at random, or played to all individual clams?

5) The details of equipment specifications are critical in this study, in my opinion. For example, the running voltage of the aquarium pump, the sound speakers working frequency in Hz, etc… As these parameters could affect the water particles uniquely.

6) On the stimulus presentation for water flow, can the authors clarify the position of hose facing the clam? For example, was the hose facing near one of the siphonal openings, or a general central position, or random positions every time?

7) On the velocity measurements (Lines 128-130), can the authors clarify further if the measurement was taken at the point where jet exits the nozzle of pump or 15 cm away from nozzle? The released jet will dampen in velocity with distance, hence, the velocity reported needs to account for this reduction. This is surely more difficult to handle in the field.

8) Line 156: For the no stimuli presentation (i.e. controls), were the clams introduced/shown the hand-held devices, but switched off during presentation? It is not clear based on this line, as it could just be describing the types of treatments. This is particularly critical to eliminate the effects of ‘equipment’ influencing the giant clam’s mantle behaviour.

9) Lines 158-160: Rather than ‘acclimate’, perhaps the authors can refer to the clams being subjected to the stimuli? The former suggests getting used to the conditions, but it is not the case.

10) Since the study is likely conducted in the field, did researchers account for disturbance by fishes or macroinvertebrates? I note that mantle twitches were observed, but from past observations, these twitches could be due to amphipods running across the mantle. What does the mantle twitches signify here?

11) More details needed on the post-stimulus disturbance are required, such as which part of the mantle had contact with the eraser, and was the region consistent across all the clams? From experience, not all clams will respond to being disturbed, and depending on the intensity and duration, it can affect the latency measured.

12) On the latency of re-emergence of mantle, I’m not quite sure if I understood the equation and number of measurements. How did the researchers obtain 103 measurements, if there were 32 samples (multiply by 4 treatments)?

Validity of the findings

As I still have questions regarding the experimental designs, I am unable to accurately assess the validity of the findings. My comments mainly deal with the phrasing of Results and Discussions.

Line 222: I think it is misleading to say “increased the number of retractions” as the actual measure was number of mantle twitches. Suggest to use “higher number of twitches recorded…” instead.

Lines 229-244: The results in the table presented are post-processed data. Would it be possible for the authors to state down the actual timings (i.e latency) for ease of comparisons?

Lines 254-256: Yes, but it should be noted that mantle retraction as an anti-predator response is a complete and sudden valve closure. The study here measured mantle twitches and partial retractions, which do not necessary conform to an anti-predator response.

Lines 256-259: Any reasons to explain why anthropogenic noise such as motorboats distract the invertebrates? It will be good to add 1-2 lines to explain the rationale based on previous studies.

Lines 265, 307: What could this unisensory receptor be, in this case, for the giant clam?

Lines 266-268: Does this study refer to multiple stimuli distracting prey? As the former sentence implies the additive effect of 2 stimuli, rather than 1 stimulus. Similar to an earlier comment, has there been other studies showing that multiple stimuli rather than single stimulus distract animals, and why? It will be good to add 1-2 lines to explain the rationale based on previous studies.

Lines 279-280: Suggest to include references that show giant clams filter-feeding for nutrients.

Lines 272-289: I’m not convinced that the giant clams re-emerge more quickly due to the need for filter-feeding, but rather to extend its outer mantle for photosynthesis. The latter has been said to contribute to almost 90% of nutrients for the giant clams, and the remainder derives from filter-feeding. Hence, the role of filter-feeding is minimal relative to the need to allow their photosymbionts to photosynthesise. There are several research papers that have confirmed this interaction.

Line 293: I’m not very convinced that the study looked at anti-predator behaviour, as the stimuli presented do not well-represent potential predators.

Lines 293-299: Can the authors expand on how these anthropogenic noises affects the animals, for example, give some specific examples.

Lines 306-308: Can the authors elaborate more on the mode of sensory in giant clams? Like what organs/receptors and how do they detect stimulus? Please indicate references.

Lines 314-332: While I acknowledge the lack of information on the topic in invertebrate species, but I suggest that the authors at least keep the discussion to marine organisms for relevant context. The transmission and detection of stimuli in terrestrial environments contrast greatly with the marine environments, therefore, it is best to remove the terrestrial examples.

Line 335: A potential anti-predator behaviour is more precise.

Additional comments

Thank you for this intriguing work! It is great to read new autecology work for the giant clams, given the major assumption that these animals move very little. Overall, I have great reservations on the experimental design, thus the outcomes of the study may not be representative of the actual treatments. I hope that the authors can supplement more information regarding their experimental process (e.g. to include an informative illustration on how the study was conducted) and clarifications of the steps, so that a better assessment of the results can be made. A more general comment is the mantle retractions here was definitively placed as an anti-predator behaviour, but it is not precise as the behaviour can also refer to others such as clearance of debris and sediments, or macro-fauna present on their mantles. While the study is designed with anti-predator avoidance, the context should be broadened to acknowledge the other possibilities.

·

Basic reporting

Raw data (i.e., actual data of frequency of mantle retractions, latency to close, and latency to re-open) should be reported as supplementary material. Please see attachment for additional comments and suggestions.

Experimental design

Some methods should be described in more details, for instance, how long were the video recordings, what time was the video recording done, how far were the giant clam individuals from each other. Please see attachment for additional comments and suggestions.

Validity of the findings

No comment.

Reviewer 3 ·

Basic reporting

no comment

Experimental design

no comment

Validity of the findings

no comment

Additional comments

This study explores the effect of exposure to two stimuli – sound and flow – on risk assessment in the giant clam Tridacna maxima. The authors provide evidence for distraction (i.e., increased response time measured as latency to close) in clams exposed to both stimuli simultaneously suggesting that giant clams perceive these stimuli in an additive fashion. The authors also show that exposure to sound and flow (alone or in combination) led to increased frequency of mantle retractions. Overall, the paper is extremely well-written, methodologically sound, and data are presented in clear and appropriate detail. Taken at face value, the results are important and are worth publishing. However, after careful revision, I have identified two topics that I feel require further discussion or clarification. The first is in regards to the assertion of the palps as the sole sensory modality in detection of the stimuli used in this study. The second deals with the potential role of habituation. In addition to these two general comments, there are a few other, minor, issues (e.g., a need for expanded citation of relevant literature) that I list below. I hope my comments are useful to help improve the manuscript.

General Comments:
(1) Single Sensory modality - Can the authors supply a reference for the primary role of the palps in detection of particle motion in giant clams? There is a previous reference to Nedelec et al. 2016 as the source for this assertion (line 63), but there appears to be no mention of the palps as a sensory organ in this study. Documentation of this role is important as it underpins the author’s conclusion that the examined stimuli are additive and thus distractive to the clams. In addition, several studies have shown there is a strong dependence on visual cues in giant clams to avoid predation by fish (Wilkens 1986) and that particle motion and pressure stimuli do not elicit a response when these stimuli originate below the plane of vision (Stasek 1965).

Wilkens LA (1986) The visual system of the giant clam Tridacna: behavioral adaptations. Biol Bull 170:393–408.
Stasek CR (1965) Behavioural adaptation of the giant clam Tridacna maxima to the presence of grazing fishes. Veliger 8:29–35

Further, Soo and Todd (2014) describe a ‘sight reaction’ in tridacnids in which clams sense and respond to particulate movement with their pinhole eyes. This response is independent from other ‘shadow responses’ in that it is not dependent on changes in illumination. Taken together, these data seem to suggest that mechanical stimuli may not be the primary means of sensation in giant clams. This could have important implications for the author’s assertion that giant clams uniquely discriminate the investigated stimuli through a single sensor – the palps (lines 35 – 36).

Soo P, Todd PA (2014) The behaviour of giant clams (Bivalvia: Cardiidae: Tridacninae). Mar Biol 161, 2699–2717. https://doi.org/10.1007/s00227-014-2545-0

(2) The second topic that I feel the authors should expand upon is the potential for habituation to stimuli in tridacnid clams. Giant clams have been shown to exhibit reduced responses when exposed to repetitive visual stimuli suggesting they are capable of habituation (Fankboner 1981). This is also well-documented in the aquarium hobby where giant clams will quickly cease to exhibit both the ‘shadow response’ and ‘sight reaction’ if regularly exposed to nearby movement or passing shadows. The authors also cite their own previous work which includes investigation of the potential for habituation in tridacnids (Dehaudt et al. 2019). Given the close proximity of the Gump research dock to the study site, is it possible that clams used in this study had habituated to motor noise/boat wakes? In this case habituation to frequent nearby boat traffic could manifest similarly to behavioral “distraction” – i.e., reduced (or slowed) response to the multimodal sound/flow stimuli as observed in this study.

Fankboner PV (1981) Siphonal eyes of giant clams and their relationship to adjacent zooxanthellae. Veliger 23:245–249

The authors do account for repeated measures in their statistical analyses, but did the authors observe any trends in latency data as a function of number of trials conducted on an individual? Similarly, could significant differences between individuals (as reported for most response variables) be explained by proximity to boat traffic (i.e., distance from the boat channel) or other potential sources of habitual, repeated stimuli?

Specific Comments/Suggestions:
Title: Although an issue of personal preference, it is generally considered preferable that titles be explanatory rather than interrogatory. I suggest altering the title to more explicitly state the results of the study, especially in regards to the specific stimuli examined.
Line 48: Check extraneous dashes aren’t present in final copy.
Lines 56 – 59: Although I agree that the list of literature regarding tridacnid behavior is, in general, sparse, the authors’ list of references seems incomplete. Authors should appropriately cite the works I’ve referred to above (all of which summarize a suite of anti-predator behaviors in tridacnid clams) as well as work by Neo and Todd (2011) who discuss visual responses in tridacnids.

Neo ML, Todd PA (2011) Quantification of water squirting by juvenile fluted giant clams (Tridacna squamosa L.). J Ethol 29:85–91

Lines 60 – 62: Alteration of anti-predator behavior in response to chemical cues has also been shown in tridacnids specifically. See the work of Neo and Todd (2011) in which both latent defense (i.e., shell characteristics) and behavior (i.e., vertical movement) were altered in Tridacna squamosa in response to a predator cue.

Neo ML, Todd PA (2011) Predator-induced changes in fluted giant clam (Tridacna squamosa) shell morphology. J Exp Mar Biol Ecol 397:21–26

Lines 75 – 75: Sentence fragment needs revising.
Line 82: Replace second instance of “changes” with “change”.
Lines 94 - 96: Change “while” to “whereas”. Add a “,” between multimodal and giant clams. Remove second period at line 96.
Lines 112 – 113: Although not published under peer review (and therefore potentially not relevant for citation here), the authors may be personally interested in data from a recent predation-response study in the same population of giant clams in Mo’orea:
Boville E (2020) Look out: giant clam (Tridacna maxima) defense response to visual stimuli in Mo'orea, French Polynesia

---

## Round 0.2 · Minor Revisions

Thank you for the hard work on the revision. With a few more edits and comments to address, I anticipate being able to accept this work in the next round. Please in particular pay attention to the comments on the Materials and Methods.

Reviewer 1 ·

Basic reporting

No further comments.

Experimental design

An illustration to explain how the experiment was conducted will be useful. This has not been addressed in this revision.

Line 119: Please clarify what dimension does the diameter of the shell refer to? Is it the width of the entire animal?

Lines 141-142: Are these five sounds similar in pitch, amplitude, etc? While the sounds were randomly introduced, it would be good for Authors to add this detail on what makes these sounds different, or similar?

Lines 187-196: Thank you for clarifying that researchers were in-water (>1m away from the test clams) during the experiments. I strongly recommend Authors include these details here, as well as to explain how they had added a 60s buffer to acclimate giant clams to their presence.

Linea 181-182: Please add detail that these 16 events were taken out and not analysed subsequently.

Validity of the findings

Lines 329-347: Thank you for clarifying on that multi-attribute studies on marine organisms are limited. This should be mentioned in the final paragraph. As much as terrestrial examples are useful, but it’ll be better for Authors to try and summarise these findings and not overwhelm the narrative that multi-attribute studies on marine organisms.

Additional comments

Thank you for the revisions and responses to my comments. I have a few more comments, but overall, well done on the revisions!

---

## Round 0.3 · accepted · Accept

I am happy to move this paper into production; thank you for the final small revisions. I look forward to seeing this paper published.